# Effect of Different Synchronization Regimens on Reproductive Variables of Crossbred (Swamp × Riverine) Nulliparous and Multiparous Buffaloes during Peak and Low Breeding Seasons

**DOI:** 10.3390/ani12040415

**Published:** 2022-02-09

**Authors:** Adili Abulaiti, Zahid Naseer, Zulfiqar Ahmed, Dong Wang, Guohua Hua, Liguo Yang

**Affiliations:** 1Key Laboratory of Animal Genetics, Breeding and Reproduction, Ministry of Education, College of Animal Science and Technology, Huazhong Agricultural University, Wuhan 430070, China; adiliabulaiti@webmail.hzau.edu.cn (A.A.); zlfqr_abbasi@yahoo.com (Z.A.); 2International Joint Research Centre for Animal Genetics, Breeding and Reproduction, Wuhan 430070, China; 3Hubei Province’s Engineering Research Centre in Buffalo Breeding and Products, Wuhan 430070, China; 4Theriogenology Section, Department of Clinical Studies, Faculty of Veterinary and Animal Sciences, Pir Mehr Ali Shah Arid Agriculture University, Rawalpindi 46000, Pakistan; zahidnaseer@uaar.edu.pk; 5Institute of Animal Science, Chinese Academy of Agricultural Sciences, Beijing 100193, China; dwangcn2002@vip.sina.com

**Keywords:** GPGMH, Ovsynch, nulliparous, multiparous, crossbred buffaloes, seasons

## Abstract

**Simple Summary:**

Parity and season are two prominent variants which influence buffalo reproductive performance and synchronization outcome. The present study compared modified new Ovsynch synchronization (GPGMH) protocol with conventional Ovsynch (OVS) in reproductive parameters of buffaloes with different parity (nulliparous and multiparous) in different seasons (peak and low breeding seasons). Compared to the conventional Ovsynch, the modified GPGMH protocol was more effective to enhance the estrus rate, ovulation rate and pregnancy rate in multiparous crossbred buffaloes. In addition, the application of GPGMH protocol improve the reproductive performance during the peak breeding season than low breeding season.

**Abstract:**

The present study was conducted to examine the effect of conventional the Ovsynch protocol (OVS) and a modified Ovsynch synchronization (GPGMH) protocol on the follicular dynamics, estrus, ovulation, and pregnancy in nulliparous and multiparous crossbred (swamp × riverine) buffaloes during different seasons. GPGMH or OVS protocols were used to synchronize nulliparous (*n* = 128; GPGMH = 94, OVS = 34) and multiparous (*n* = 154; GPGMH = 122, OVS = 32) buffaloes during the peak (*n* = 186; GPGMH = 143, OVS = 43) and low breeding (*n* = 96; GPGMH = 73, OVS = 23) seasons. Buffaloes were monitored for follicular dynamics, estrus response, ovulation, and pregnancy rates. The results showed that protocol, parity, and season had significant effects on estrus, ovulation, and pregnancy variables, and interactions among parity and protocol, season and protocol, and season and parity were observed for few of reproductive indices in the crossbred buffaloes. There were no significant (*p* > 0.05) interaction for protocol, parity and season. In multiparous buffaloes, the application of the GPGMH protocol significantly (*p* < 0.05) increased the interaction to the interval to estrus onset after the second GnRH, estrus response, ovulation rate, and pregnancy rate, and lowered (*p* < 0.05) the silent estrus when compared with the conventional OVS protocol. During the peak breeding season, the application of the GPGMH protocol significantly (*p* < 0.05) improved the interaction to the estrus response, ovulation rate, and pregnancy rate, while it lowered (*p* < 0.05) the silent estrus incidence when compared to the conventional OVS protocol. In conclusion, the GPGMH protocol, in comparison to the OVS protocol, improves the follicular dynamics, estrus response, ovulation, and pregnancy rates in crossbred multiparous buffaloes during the peak breeding seasons.

## 1. Introduction

Buffalo productivity is influenced by the seasonality pattern of breeding. The delayed attainment of puberty in heifers and prolonged inter-calving intervals in multiparous buffaloes are major determinants for low fertility [1,2]. Poor ovarian activity or short estrus duration with subtle estrus signs exaggerate the conditions during the summer season [1,3,4]. Knowledge gaps in follicular dynamics associated with seasonal breeding behavior in nulliparous and multiparous buffaloes are a major limiting factor and a better understanding could help to manipulate the cyclicity and breeding patterns [5].

In recent decades, several estrus synchronization protocols have been applied aiming to manipulate ovarian activity in nulliparous and multiparous buffaloes during peak and low breeding seasons [6]. Synchronization protocols are used in buffaloes to breed around the year to decrease the anestrous incidence, particularly in summer. To reduce the incidence of anestrous [7] and embryonic mortality [8] and enhance the use of AI [9,10], the ovulation synchronization protocols were also implemented in buffaloes for maximizing pregnancy rates per AI [11,12]. In general, the ovulation synchronization protocols did not provide the optimal fertility results during the low breeding season when a limited number of cyclic buffaloes were available [13]. The incorporation of LH/hCG [14,15], estradiol [16], eCG [17], PGF_2_α [18,19], progesterone [11,20], insulin [21], and an inhibin inhibitor [22] in the OVS protocol has been tested in maximizing pregnancy per AI in buffaloes during breeding and low breeding seasons.

Decreased progesterone concentrations at the time of AI and greater CL volume are prerequisites for future pregnancy establishment in buffaloes [23]. Our previous study showed that the inclusion of mifepristone and hCG in the modified OVS protocol (GPGMH) could improve the fertility by enhancing the ovulation rates and subsequent conception rate in tested buffaloes [24]. Mifepristone is a kind of progesterone (P4) receptor antagonist that works as an antiprogestogen by blocking the progesterone receptor, in turn rapidly reducing the P4 level, and further promoting the luteinizing hormone (LH) surge for ovulation [25]. Human chorionic gonadotropin (hCG) is a hormone produced by the human placenta after embryo implantation, which interacts with its receptor in the ovary and promotes the corpus luteum (CL) to secrete more P4 during the first trimester for pregnancy maintenance [26]. However, the success ratio of OVS in buffaloes is variable in nulliparous and multiparous buffaloes during peak and low breeding seasons; therefore, it is necessary to test the modified OVS protocol to overcome the issues of seasonality and parity in the buffalo production system. The present study aimed to determine the effect of OVS and GPGMH protocols on follicular dynamics, estrus expression, ovulation occurrence, and pregnancy outcomes in nulliparous and multiparous crossbred (swamp × riverine) buffaloes during different seasons.

## 2. Materials and Methods

### 2.1. Animal Ethics

The study was approved by the tab of animal experimental ethical inspection of the Laboratory Animal Centre, Huazhong Agriculture University (Approval ID: HZAUBU-2017-001). All experimental protocols were performed in accordance with the guidelines of the Committee of the Animal Research Institute, Huazhong Agricultural University, China.

### 2.2. Description of Experimental Animals and Husbandry Practices

In the present study, cyclic nulliparous (*n* = 128) and multiparous (*n* = 154) crossbred buffaloes (Nili Ravi × Jiangha; *Bubalus bubalis*), maintained at a private buffalo farm in Hubei Jinniu Co., Ltd., Hubei province, China, during peak breeding (September to April, average ambient temperature: 5–28 °C, relative humidity: 55–75%, and temperature-humidity index, THI: 67) and low breeding (May–August, 35–39 °C, RH: 80–90%, and THI: 88) seasons. The nulliparous buffaloes varied from 2.5 to 3.5 years old, with a moderate body condition score (BCS 2–3.5) and average body weight (BW) 433 ± 72 kg, whereas selected multiparous buffaloes had 2.5–3.5 BCS, 5–8 years of age, and 588 ± 89 kg BW at the beginning of the experiment.

All buffaloes were fed with total mixed rations (TMR) (8% corn silage, 16% peanut vine, 17% soy pulp, 2% rice straw, 38% corn, 16% soybean meal, 6.0% linen, 6% cottonseed cake, 17.5% corn meal, 10% vinasse, 0.5% sodium bicarbonate, and 6% premixed material). Machine milking was in practice twice a day (6:00 and 18:00). Ad libitum fresh and clean water was provided routinely. Buffaloes were housed in an open shed with a cemented roof top and a two-sided wall fenced by galvanized wire mesh. The multiparous buffaloes were clinically and physically healthy with a normal calving history and reproductive soundness. Additionally, the nulliparous buffaloes were also screened through ultrasonography on a monthly basis to monitor cyclicity or any reproductive anomaly prior to the start of the trial.

### 2.3. Estrus Synchronization Protocols

In this experiment, GPGMH protocol was shown as Figure 1 [24]). Animals were muscularly injected with the first GnRH (gonadotropin-releasing hormone, 400 μg on day 0, PGF_2_α (prostaglandin F_2_α, 0.5 mg) on day 7, the 2nd GnRH and mifepristone (0.4 mg/kg) simultaneously on day 9, and hCG (human chorionic gonadotropin, 2000IU,) on day 5 after AI. OVS protocol was used as described by (1st GnRH at day 0, PGF_2_α at day 7, and 2nd GnRH on day 9) [27].

A total of two hundred and eighty-two nulliparous (*n* = 128) and multiparous (*n* = 154) buffaloes were selected. The selected buffaloes were synchronized through OVS (*n* = 66) and GPGMH (*n* = 216) protocols, keeping the categories of parity (nulliparous (*n* = 128; GPGMH = 94, OVS = 34) and multiparous (*n* = 154; GPGMH = 122, OVS = 32)) and seasons (peak breeding seasons (*n* = 186; GPGMH = 143, OVS = 43), low breeding seasons (*n* = 96; GPGMH = 73, OVS = 23)).

### 2.4. Follicular Dynamics, Estrus Detection, AI, and Pregnancy Diagnosis

The ovaries were scanned transrectally using an ultrasound machine (WED-9618-v, equipped with LV2-3/6.5 MHz rectal probe, Shenzhen Well.D Medical Electronics Co., Ltd., Guangdong, China), starting from day 6 to day 12 of the protocol for follicular dynamics. Ultrasonography was performed twice a day and follicular dimensions were calculated using follicle diameter recordings at different intervals. The ovulation was defined as the time of the sudden disappearance of dominant follicles (>10 mm) following the 2nd GnRH treatment. Ovulatory follicle diameter was recorded on the basis of reciprocal measurements from ovulation to the initial follicular scan. Anovular follicles were defined as the presence of a follicle (>15 mm) persisting from the last scan at day 12 of the protocol.

Estrus response was detected by daily visual observations (edematous vulva, presence of cervical mucus, being sniffed by other female mates, or the presence of a >9 mm follicle with increased uterine tone). Buffaloes were inseminated at a fixed-time AI schedule (16 h after the 2nd GnRH dose) by a skilled AI technician using frozen-thawed semen. Pregnancy diagnosis was confirmed by transrectal ultrasonography at day 40 post AI and confirmed by the presence of amniotic fluid with a viable embryo.

### 2.5. Statistical Analyses

Data were analyzed using statistical software (SPSS version 16.0, IBM Corp., Armonk, NY, USA). The values are expressed as mean ± SD. Repeated measures analysis of variance (ANOVA) was used to compare the effect of season, protocols, and parity for follicular diameters and durations to estrus or ovulation and pregnancy rate. In addition, the influence of parity status, protocols, and season on the estrus response, silent estrus, ovulation rate, follicular cyst incidence, and pregnancy rate were compared using logistic regression with the model consisting of parity (multiparous or nulliparous), season (breeding seasons, non-breeding seasons), and treatment (GPGMH or OVS) as main factors and their interactions. The values *p* < 0.05 was considered as statistically significant.

## 3. Results

### 3.1. Effect of Parity, Protocol, and Breeding Season on the Reproductive Performance in Crossbred Buffaloes

Table 1 shows the main effects and interactions of protocol, parity, and season on different reproductive indices for the buffaloes. Significant effects (*p* < 0.05) of different protocols on the follicle diameter at day 6, interval to estrus after the second GnRH, estrus response, estrus duration, silent estrus rate, interval to ovulation after the second GnRH, ovulation rate, and pregnancy rate were observed. There was a significant effect of parity on the follicle diameter at day 6, estrus response, estrus duration, interval to ovulation after the second GnRH, ovulation follicle diameter, ovulation, and pregnancy rate in buffaloes. Also, a significant effect (*p* < 0.05) of season on the interval to estrus or ovulation after the second GnRH, estrus response, and ovulation and pregnancy rate was observed.

Protocol and parity showed strong interactions (*p* < 0.05) for the interval to estrus after the second GnRH, estrus response, silent estrus rate, interval to ovulation after the second GnRH, ovulation follicle diameter, and ovulation rate in buffaloes. Significant interactions (*p* < 0.05) between season and protocol were observed in relation to the interval to estrus after the second GnRH, estrus response, silent estrus rate, ovulation rate, and pregnancy rate in the crossbred buffaloes. In addition, season and parity interacted significantly (*p* < 0.05) for ovulation rate in the crossbred buffaloes. However, there were no significant interactions for protocol, parity, and season for any reproductive variable in the buffaloes (Table 1).

### 3.2. Effect of Synchronization Protocols on the Reproductive Parameters of Different Parities of Buffaloes

The previous study showed strong interactions between parity and protocols. To further identify the effect of protocols for different parities, we compared the GPGMH and conventional Ovsynch protocol effects on the reproductive parameters mentioned above. In multiparous buffaloes, the application of the GPGMH protocol significantly (*p* < 0.05) increased the interaction with the interval to estrus onset after the second GnRH, estrus response, ovulation rate, and pregnancy rate and decreased the silent estrus rate (*p* < 0.05) when compared with the conventional OVS protocol. However, the application of GPGMH did not promote reproductive performance in nulliparous buffaloes. The pregnancy rate was influenced by parity (multiparous) and protocols (GPGMH), but no interaction was found between the parity and used protocols (Table 2).

The estrus duration, interval to ovulation after the second GnRH, ovulatory follicle diameter, and incidence of follicular cysts were similar between the GPGMH and OVS protocols in multiparous and nulliparous buffaloes. In addition, no interaction (*p* < 0.05) was found between the influencing factors parity and protocol for follicle diameter at day 6, estrus duration, or the incidence of follicular cysts (Table 2).

### 3.3. Effect of Protocols on the Reproductive Performance of Buffaloes in Different Seasons

The effects of seasons and synchronization protocols on the reproductive indices of the synchronized crossbred buffaloes are presented in Table 3. The previous study showed strong interactions between seasons and protocols. To further identify the effect of protocols on different seasons, we compared the GPGMH protocol and conventional Ovsynch effects on the reproductive parameters mentioned above. During the peak breeding season, the application of the GPGMH protocol significantly increased (*p* < 0.05) the interaction to the estrus response, ovulation rate, and pregnancy rate, and decreased (*p* < 0.05) the silent estrus rate when compared with the conventional OVS protocol. However, the application of GPGMH did not promote reproductive performance during the low breeding seasons. The pregnancy rates were influenced by season (PBS) and protocol (GPGMH) with a significant interaction (*p* < 0.05) between the influencing factors (seasons and protocols).

The follicle diameter at first observation, interval to estrus onset after second GnRH, and incidence of follicular cysts did not vary among GPGMH and OVS protocols during the peak and low breeding seasons. In addition, no interaction (*p* < 0.05) was found between the influencing factors parity and protocol for follicle diameter at day 6, estrus response, estrus duration, interval to ovulation after the second GnRH, ovulation follicle diameter, or the incidence of follicular cysts (Table 3).

## 4. Discussion

The hypothesis of the present experiment was based on a preliminary study conducted [24] using crossbred buffaloes which reported improved reproductive parameters when hCG and mifepristone were included in the conventional OVS protocol. In the present study, nulli- and multiparous buffaloes were synchronized by the GPGMH and OVS protocol during different seasons. The observed findings indicate that the GPGMH protocol seems advantageous over OVS for some of the fertility variables in nulli- and multiparous buffaloes irrespective of the season. The GPGMH enhanced the estrus, ovulation, and pregnancy rates comparatively in the crossbred buffaloes during the peak and low breeding seasons. Previously, many reports illustrated the effectiveness of a modified Ovsynch protocol in buffaloes and reported the significant improvement in pregnancy per AI compared to the conventional Ovsynch protocol [11,26] with a few exceptions [17,18,19,21,28,29]. The data of the current study could be valuable information to apply in heifers and adult buffaloes to reduce the age at first calving and the post-partum calving interval, which are the main reproductive issues in buffalo herds.

Buffaloes require higher energy for milk production and calf rearing during the early and mid-lactation period, and spare less energy for the reproductive process; therefore, multiparous buffalo respond inadequately when any synchronization protocol is employed. It is also the general opinion that buffalo heifers respond well against the synchronization compared to adult/multiparous buffaloes, but sub-optimum body conditions of heifers could affect the fertility response [12]. In addition, follicular and luteal activity and nutritional status following breeding are important pre-determinants for the success of any synchronization protocol. In the present study, buffalo heifers’ reproductive variables against GPGMH or OVS protocols were not desirable as observed previously [25,26]. However, improved estrus and ovulation variables were observed in GPGMH treated multiparous buffaloes compared to the nulliparous buffaloes. Also, according to previous reports [15,16,17,18,19,20,21,22], buffaloes exhibited better fertility outcomes when synchronized through OVS by incorporating hCG, estradiol, eCG, PGF2a insulin, and an inhibin inhibitor. Improved ovulation parameters in the present study are possibly associated to the use of mifepristone at the time of the second GnRH, which increased the estrus expression and ovulation rate by lowering the progesterone effect.

Furthermore, estrus parameters such as the estrus expression rate, duration of the estrus stage, and onset of estrus after the second GnRH showed a variability in response to protocol and parity. Shorter estrus duration in nulliparous buffaloes synchronized with the OVS protocol might be due smaller ovulation follicle size. In addition, the ineffectiveness of PGF2α in the OVS protocol could also be a contributory factor to low estrus response in nulliparous buffaloes. The current data showed that the estrous duration of female buffaloes synchronized through GPGMH or OVS was shorter (~14 to 16 h) compared to the estrous duration (~18 h) of swamp buffaloes synchronized through OVS [30]; this variation might be due to breed difference. The incidence of silent estrus is a major issue in buffaloes during different estrous synchronization programs. A large proportion (30%) of buffaloes ovulated without exhibiting estrus signs after a PGF2α-based synchronization regimen. The use of PGF2α in a synchronization protocol ensures the reduced concentration of progesterone in plasma in cyclic buffaloes [31]. No expression of estrus signs following PGF2α treatment explained the ineffectiveness of PGF2α in such circumstances. The use of mifepristone appeared to be advantageous due to the antiprogestogenic mechanism that reduces progesterone concentration. Also, the low incidence of silent estrus in the GPGMH treatment (12%) group compared to the OVS group (22%) might be associated with the use of mifepristone at the time of the second GnRH of the protocol, which reduced the action of the progesterone hormone by occupying the receptors. In future, the estimation of the progesterone hormone level following mifepristone treatment, at the time of breeding/AI, and prior to/after ovulation would enhance the understanding of silent estrus physiology in buffaloes.

Another plausible reason for the success of GPGMH in multiparous buffaloes could be the use of hCG post-AI. Under such conditions, hCG, seeming to act as a substitute of LH as it haves the biological activity of LH, was used at the time of AI or 5 days later to promote the accessory CLs to maintain an optimum progesterone level. Because of its long-standing action in circulation, its dissociation from the LH receptor is gradual, which helps the granulosa cells to transform into luteal cells. It has been reported that CL formed after hCG use has a similar steroidogenic activity on the progesterone hormone [26]. Additionally, its usage increases the functional CL area for the optimum progesterone level [24]. Also, the low response of heifers to GPGMH or OVS compared to multiparous buffaloes could be the result of the use of an inappropriate body weight of buffalo heifers or stress occurring during rigorous rectal palpation for ovarian dynamics, which led to low follicular output. In contrast, Derar et al. [32] reported higher results in heifers than in adult buffaloes by synchronizing through the OVS protocol. Therefore, the achieved reproductive variables in multiparous buffaloes after the GPGMH protocol indicate that the application of GPGMH could be beneficial to minimize the calving interval successfully.

Although, the buffalo is a polyestrous animal, a seasonality pattern in breeding has been reported [12,18,28,29] in different buffalo breeds, and the summer season is categorized as a low breeding season. It has been observed that 30% of buffalo experienced post-partum anestrous and 70% showed estrous activity during the first 90 days of calving. During the summer season, the majority of cyclic buffaloes also undergo the state of anestrous because of underfeeding, high ambient temperatures or humidity, poor husbandry practices, and some element of a long photoperiod [33]. Postpartum anestrous in buffaloes continues if it does not breed in the current breeding season, and anestrous will most likely persist in the herds until the next breeding season, and thus the days open are reduced and yearly calving is not possible as is practiced in cattle. Estrus synchronization protocols in buffaloes provide a layout to overcome the less marked estrus behavior by applying the timed-AI breeding program in herds. This strategy helps to shorten the post-partum interval in buffaloes during low and peak breeding seasons. The use of an optimum synchronization protocol considering the season and buffalo cyclicity could lead to promising fertility. In the present study, the MOVS protocol was applied, keeping in view the cyclicity and seasonal issues. In the present study, MOVS was a beneficial regimen to improve the estrus response and ovulation during peak (autumn) and low breeding seasons (winter, spring, summer). The OVS treated buffaloes showed a low estrus response with a reduced ovulation rate and ovulatory follicle size, particularly during the low breeding season (summer), which reflects the effects of high ambient temperature and humidity in the study site. It is also a general opinion that the application of ovulation synchronization protocols in buffaloes result in lower fertility variables because of the stressful conditions of the summer season [33]. However, modification of the protocol is a choice for the successful application of an OVS regimen in buffalo herds. Similar to the present study, Carvalho et al. [25,27] synchronized the buffaloes through a modified OVS protocol during the low breeding season and obtained better fertility. In future, the application of GPGMH using larger buffalo herds could better explain the fertility traits in terms of pregnancy outcome. However, the use of GPGMH seems an effective alternative method to the traditional OVS protocol for the improvement of some reproductive traits in multiparous buffaloes throughout the year.

## 5. Conclusions

In conclusion, the GPGMH protocol enhances the estrus, ovulation, and pregnancy traits in multiparous buffaloes. Moreover, the modified Ovsynch protocol (GPGMH) could be applicable during peak and low breeding seasons for promising reproductive results in crossbred multiparous buffaloes.

## Figures and Tables

**Figure 1 animals-12-00415-f001:**
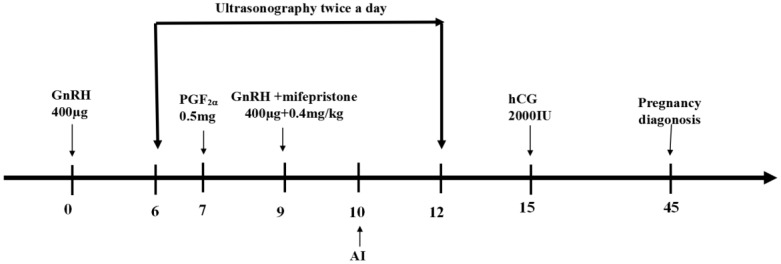
The schematic description of the modified Ovsynch protocol (GPGMH) applied in nulliparous and multiparous buffaloes during peak and low breeding seasons [24].

**Table 1 animals-12-00415-t001:** Effect of protocol (GPGMH vs. OVS), parity (nulliparous vs. multiparous), season (peak breeding season (PBS) vs. low breeding season (LBS)), and their interactions for different reproductive parameters of crossbred buffaloes.

Variables	Protocol		Parity		Season		Main Effects			Interactions			
	GPGMH (*n* = 216)	OVS (*n* = 66)	Multiparous (*n* = 154)	Nulliparous (*n* = 128)	PBS (*n* = 186)	LBS (*n* = 96)	Protocol	Parity	Season	Protocol × Parity	Protocol × Season	Parity × Season	Protocol × Parity × Season
Follicle diameter at day 6 (mm)	7.3 ± 2.3	8.2 ± 2.6	8.6 ± 2.6	6.9 ± 1.9	7.4 ± 0.9	7.5 ± 0.9	0.020	<0.0001	0.5705	0.290	0.327	0.617	0.249
Interval to estrus after 2nd GnRH (h)	9.7 ± 1.1	8.5 ± 1.2	9.1 ± 1.1	9.2 ± 1.2	8.9 ± 0.7	9.3 ± 0.7	<0.0001	0.722	0.0500	0.043	0.028	0.402	0.548
Estrus response (%)	186/216 (86.1)	50/66 (75.8)	139/154 (89.0)	97/128 (75.8)	162/186 (87.1)	74/96 (77.1)	0.0463	0.022	0.0105	0.042	0.014	0.156	0.134
Estrus duration (h)	15.7 ± 1.3	14.1 ± 1.7	16.0 ± 0.9	14.9 ± 1.6	15.0 ± 0.6	15.1 ± 0.5	0.036	<0.0001	0.7070	0.716	0.822	0.461	0.252
Silent estrus rate (%)	35/216 (16.2)	19/66 (28.8)	26/154 (16.9)	28/128 (21.9)	28/186 (15.1)	26/96 (27.1)	0.0017	0.084	0.2780	0.024	0.031	0.625	0.169
Interval to ovulation after 2nd GnRH (h)	24.4 ± 0.9	23.8 ± 0.9	24.4 ± 0.9	23.7 ± 0.9	24.3 ± 0.5	23.9 ± 0.4	0.0002	<0.0001	0.0106	0.047	0.765	0.075	0.164
Ovulation follicle diameter (mm)	13.1 ± 2.3	13.0 ± 1.8	13.9 ± 1.8	12.1 ± 1.7	13.7 ± 0.8	13.3 ± 0.8	0.898	<0.0001	0.0938	0.051	0.879	0.467	0.392
Ovulation rate (%)	177/216 (81.9)	43/66 (65.2)	130/154 (84.4)	90/128 (70.3)	154/186 (82.7)	66/96 (68.8)	0.0377	0.0044	0.0012	0.053	0.013	0.022	0.149
Incidence of follicular cysts (%)	5/216 (2.3)	2/66 (3.0)	4/154 (2.6)	3/128 (2.3)	4/186 (2.2)	3/96 (3.1)	0.874	0.736	0.2890	0.713	0.790	0.449	0.359
Pregnancy rate (%)	98/216 (45.4)	21/66 (31.8)	73/154 (47.4)	46/128 (35.9)	85/186 (45.7)	34/96 (35.4)	0.0511	0.0137	0.0372	0.300	0.0530	0.519	0.123

The data were presented as mean ± standard deviation. The *p*-values (*p* < 0.05) show the significance for protocol, parity, or season.

**Table 2 animals-12-00415-t002:** The follicular dynamics, estrus, ovulation, and pregnancy in nulliparous and multiparous crossbred buffaloes synchronized either by modified Ovsynch (GPGMH) or Ovsynch (OVS) protocol.

Variables	Multiparous (*n* = 154)		Nulliparous (*n* = 128)		Statistical Effect and Interaction *		
	GPGMH (*n* = 122)	OVS (*n* = 32)	GPGMH (*n* = 94)	OVS (*n* = 34)	*p*	PT	*p* × PT
Follicle diameter at day 6 (mm)	8.4 ± 2.3	8.7 ± 2.8	6.1 ± 1.6 ^B^	7.6 ± 2.2 ^A^	0.0003	0.0514	0.1774
Interval to estrus onset after 2nd GnRH (h)	9.9 ± 1.2 ^a^	8.3 ± 1.0 ^b^	9.5 ± 0.9	8.8 ± 1.4	0.7888	0.0001	0.0424
Estrus response (%)	114/122 (93.4) ^a^	25/32 (78.1) ^b^	72/94 (76.6)	25/34 (73.5)	0.0030	0.1410	0.0210
Estrus duration (h)	16.2 ± 1.2	15.7 ± 0.7	15.2 ± 1.1	14.5 ± 2.1	<0.0001	0.0350	0.6305
Silent estrus rate (%)	15/122 (12.3) ^b^	11/32 (34.4) ^a^	20/94 (21.3)	8/34 (23.5)	0.0080	0.0840	0.0100
Interval to ovulation after 2nd GnRH (h)	24.6 ± 0.9	24.3 ± 0.8	24.2 ± 0.9	23.2 ± 0.8	<0.0001	0.0027	0.0310
Ovulation follicle diameter (mm)	14.4 ± 2.2	13.5 ± 1.4	11.8 ± 1.5	12.5 ± 2.0	<0.0001	0.7839	0.0200
Ovulation rate (%)	109/122 (89.3) ^a^	21/32 (65.6) ^b^	68/94 (72.3)	22/34 (64.7)	0.0056	0.1060	0.0470
Incidence of follicular cysts (%)	3/122 (2.5)	1/32 (3.1)	2/94 (2.1)	1/34 (2.9)	0.7347	0.9018	0.9600
Pregnancy rate (%)	62/122 (50.8) ^a^	11/32 (34.4) ^b^	36/94 (38.3)	10/34 (29.4)	0.0170	0.0538	0.3330

* *p* shows the effect of parity (nulliparous vs. multiparous). PT indicates the effect of protocol (Ovsynch vs. modified Ovsynch) and *p* × PT indicates the interactions between parity and protocol. The values of *p* < 0.05 show the significance for parity or protocol. Lowercase letters (a, b) in the same row indicate significant differences (*p* < 0.05) in multiparous buffaloes, whereas uppercase letters (A, B) in the same row indicate significant differences (*p* < 0.05) in nulliparous buffaloes.

**Table 3 animals-12-00415-t003:** Effect of season [peak breeding season (PBS) vs. low breeding season (LBS)], synchronization protocol (GPGMH and OVS), and interactions on follicular dynamics, estrus, ovulation, and pregnancy in crossbred buffaloes.

Variables	PBS (*n* = 186)		LBS (*n* = 96)		Statistical Effect and Interaction *		
	GPGMH (*n* = 143)	OVS (*n* = 43)	GPGMH (*n* = 73)	OVS (*n* = 23)	S	PT	S × PT
Follicle diameter at day 6 (mm)	7.2 ± 1.3	7.5 ± 1.4	7.1 ± 1.2	7.9 ± 1.5	0.1009	0.5580	0.4664
Interval to estrus onset after 2nd GnRH (h)	9.2 ± 0.7	8.6 ± 1.0	9.1 ± 0.7	9.5 ± 1.2	0.6152	0.0410	0.0239
Estrus response (%)	129/143 (90.2) ^a^	33/43 (76.7) ^b^	57/73 (78.1)	17/23 (73.9)	0.0020	0.0069	0.0940
Estrus duration (h)	15.4 ± 0.7 ^a^	14.7 ± 0.9 ^b^	15.5 ± 0.9 ^A^	14.7 ± 0.8 ^B^	<0.0001	0.5398	0.9023
Silent estrus rate (%)	16/143 (11.2) ^b^	12/43 (27.9) ^a^	19/73 (26.0)	7/23 (30.4)	0.0795	0.6838	0.0009
Interval to ovulation after 2nd GnRH (h)	24.7 ± 0.7^a^	23.9 ± 0.6 ^b^	24.5 ± 0.7 ^A^	23.4 ± 0.5 ^B^	<0.0001	0.0067	0.3589
Ovulation follicle diameter (mm)	14.1 ± 1.1 ^a^	13.6 ± 1.3 ^b^	14.4 ± 1.0 ^A^	13.4 ± 1.1 ^B^	0.4645	0.0043	0.9656
Ovulation rate (%)	125/143 (87.4) ^a^	29/43 (67.4) ^b^	52/73 (71.2)	14/23 (60.9)	<0.0001	0.0003	0.0388
Incidence of follicular cysts (%)	3/143 (4.2)	1//43 (4.7)	02/73 (6.8)	1/23 (13.0)	0.1878	0.6794	0.3799
Pregnancy rate (%)	70/143 (49.0) ^a^	15/43 (34.9) ^b^	28/73 (38.4)	6/23 (26.1)	0.0549	<0.0001	0.0019

* S shows the effect of protocol (Ovsynch vs. modified Ovsynch). PT indicates the effect of season (breeding vs. non-breeding) and S × PT indicates the interactions between season and protocol. The values of *p* < 0.05 show significance for season or protocol. Lowercase letters (a, b) in the same row indicate significant differences (*p* < 0.05) in the peak breeding season and uppercase letters (A, B) in the same row indicate significant differences (*p* < 0.05) in the low breeding season.

## Data Availability

The authors will provide original data upon request.

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
