# Peer review of "Effect of Different Synchronization Regimens on Reproductive Variables of Crossbred (Swamp × Riverine) Nulliparous and Multiparous Buffaloes during Peak and Low Breeding Seasons"

_animals, 2022, doi:10.3390/ani12040415_

Round 1

Reviewer 1 Report

31 …  GPGMH protocol improved ovulatoration which result to the promising fertility results in crossbred nulli- and multiparous buffaloes.
Check spelling and rephrase

177 … Results showed that a significant affect (P<0.05)
Check spelling

198 … A significant affect (P
<0.05) of Season on Interval to estrus after 2nd GnRH>Check spelling

207 … rate was tended higher (P<0.05)
The use of the verb:  to tend, in my opinion is not correct and not necessary

209 … was tended lower
idem

226 … estrus incidence was tended lower (P<0.05)
idem

232 … were significantly (P<0.05) greater
Greater is not in my opinion the correct word

235 … Ovulation rate was greater
idem

238 … An interaction was tended for interval
The use of the verb:  to tend, in my opinion is not correct and not necessary

248 … and tendency between protocol and season was observed in relation to
A tendency is something that statistically is defined, and not unanimously accepted, I would avoid

329 … Estrus rate was greater
Greater is not, in my opinion, the correct word

332 … estrus duration were significantly higher
Higher is not, in my opinion, the correct word

335 … Silent estrus (rate)  significantly

338-339 … Synchronization protocol and season both of affected (P<0.05) the interval to ovulation after 2nd GnRH 339 injection with a no significant interaction (P><0.05) the interval to ovulation after 2nd GnRH injection with a no significant interaction (P>0.05) in buffaloes
I do not understand

348 There was an significant interaction (P<0.05) was><0.05) was found for ovulation rate in buffaloes between season and protocol used.
Rephrase

350 There was no significant (P<0.05) effect of season or protocol and significant interaction used on occurrence><0.05) effect of season or protocol and significant interaction used on occurrence
I do not understand

369-370-371 400
 Why authors are cited by name here?

Author Response

Response to Reviewer 1 Comments

We appreciate your valuable suggestions. We have went through the revision and tried the best to address all comments in the current revision. The revised text has been highlighted in blue font. We hope that revised manuscript would have satisfied responses for the editor and the reviewers.

Thank you,

Liguo Yang

Comments from the Editor and Reviewer:    

Reviewer 1
31 … GPGMH protocol improved ovulatoration which result to the promising fertility results in crossbred nulli- and multiparous buffaloes. Check spelling and rephrase

Author response: Thanks for your valuable comments and suggestion. The text has been revised and modified as “In addition, the application of GPGMH protocol improve the reproductive performance during the peak breeding season than low breeding season.”

177 … Results showed that a significant affect (P<0.05) Check spelling 

Author response: The text has been revised. The whole section of results has been modified to omit the typos and grammatical errors.

198 … A significant affect (P<0.05) of Season on Interval to estrus after 2nd GnRH >Check spelling

Author response: This section has been revised and modified.

207 … rate was tended higher (P<0.05) The use of the verb: to tend, in my opinion is not correct and not necessary

Author response: This text has been modified.

209 … was tended lower idem

Author response: This section has been revised.

226 … estrus incidence was tended lower (P<0.05) idem

Author response: This section has been checked and properly modified.

232 … were significantly (P<0.05) greater Greater is not in my opinion the correct word

Author response: This section has been modified.

235 … Ovulation rate was greater idem

Author response: Wrote according to observed data.

238 … An interaction was tended for interval The use of the verb: to tend, in my opinion is not correct and not necessary

Author response: This section has been revised.

248 … and tendency between protocol and season was observed in relation to A tendency is something that statistically is defined, and not unanimously accepted, I would avoid

Author response: This section has been revised.

329 … Estrus rate was greater Greater is not, in my opinion, the correct word

Author response: The results were rewritten as per observations.

332 … estrus duration were significantly higher Higher is not, in my opinion, the correct word

Author response: Revised according to data observations.

335 … Silent estrus (rate) significantly

Author response: This section has been revised and modified.

338-339 … Synchronization protocol and season both of affected (P<0.05) the interval to ovulation after 2nd GnRH injection with a no significant interaction (P><0.05) the interval to ovulation after 2nd GnRH injection with a no significant interaction (P>0.05) in buffaloes I do not understand

Author response: The part has been modified and omitted the irrelevant material.  

348 There was an significant interaction (P<0.05) was><0.05) was found for ovulation rate in buffaloes between season and protocol used. Rephrase

Author response: The text has been revised and modified .

350 There was no significant (P<0.05) effect of season or protocol and significant interaction used on occurrence><0.05) effect of season or protocol and significant interaction used on occurrence I do not understand

Author response: This section has been revised and modified .

369-370-371 400 Why authors are cited

Author response: Thanks for your valuable comments and suggestion.There was some mistakes in text, we did not quote references according to journal format. Revised text is according to instructions.

Reviewer 2 Report

Thank you for response for my comments.

I have two comments.

Please do not use the abbreviation (GPGMH) in title and write down what is the GPGMH in introduction for easily understanding for readers.

I think this kind of experiment must be the randomized controlled trial to minimized the bias of the treatment effect. However, numbers of replicate in Control and GPGMH were very different in this study (Control: n=66, GPGMH: n=216). So, my question is that how did you divided replicate to minimized the bias to achieve the randomized controlled trial?

Author Response

Response to Reviewer 2 Comments

We appreciate your valuable suggestions. We have went through the revision and tried the best to address all comments in the current revision. The revised text has been highlighted in blue font. We hope that revised manuscript would have satisfied responses for the editor and the reviewers.

Thank you,

Liguo Yang

Comments from the Editor and Reviewer:    

Reviewer 2

Please do not use the abbreviation (GPGMH) in title and write down what is the GPGMH in introduction for easily understanding for readers.

Author response: Thanks for your valuable comments and suggestion. The title has been modified as per instruction and details of GPGMH protocol were given in introduction section already.

I think this kind of experiment must be the randomized controlled trial to minimized the bias of the treatment effect. However, numbers of replicate in Control and GPGMH were very different in this study (Control: n=66, GPGMH: n=216). So, my question is that how did you divided replicate to minimized the bias to achieve the randomized controlled trial?

Author response: Thanks for your suggestion. We conducted a study previously (Abulaiti et al. GPGMH, a New Fixed Timed-AI Synchronization Regimen for Swamp and River Crossbred Buffaloes (Bubalus bubalis). Front. Vet. Sci. 2021) to observe the effect of GPGMH vs other synchronization protocol where we found the GPGMH as an advantageous protocol. In the present study, inclusion of low number of animal in OVS protocol was unavailability of animals to study the aspect of season and parity, although we tried to increase the number animals in the group even after completion of project pursuing the same observation in a previous submission.  

It is our general and experimental observation that the control group (OVS protocol) had low fertility which affect the farm economy. In buffaloes farming we were commonly using GPGMH estrus synchronization protocol. In this study OVS protocol was used as control group for comparison with GPGMH protocol. When we compare the fertility and cost of protocols, OVS always resulted low pregnancy with OVS which hindered to add the large number of animals in OVS group. Although, there is difference in animal number but it showed significance at different observations.

Round 2

Reviewer 2 Report

Thank you for your replies for my comments.

I understand the your idea and concept of you manuscript.

This manuscript is a resubmission of an earlier submission. The following is a list of the peer review reports and author responses from that submission.

Round 1

Reviewer 1 Report

Type of manuscript: Article: Manuscript ID: animals-1383684

Title: Effect of Modified and Conventional Ovsynch Regimens on Reproductive Variables of Crossbred (Swamp × Riverine) Nulliparous and Multiparous Buffaloes during Different Seasons.

The objective of the study was to compare the effect of two different methods of timed AI on follicular dynamic, estrual behavior, incidence of ovulation, and pregnancy per AI in Buffaloes across parties and seasons. 

Briefly, the manuscript is written decently. However, in the reviewer’s humble opinion, the authors have not explained certain areas well and many items require clarification. Please see the detailed comments:

L 55 Change round to “around

L56, change lowering to “decreasing

L 59: What the authors mean by “fertility level”. Please be more specific

L 63: Change maximizing fertility to maximizing pregnancy per AI

L 64-65. The authors state that the results of the modified OVS protocols are variable but “show successful induction of ovulation and subsequent CL maintenance. The authors have to be more specific. Were all the above mentions (L 61-62) protocol successful in ovulation and CL formation? OR some were and some of them were not?

L 66: What do the authors mean by “greater CL” Greater? Is it great volume? Greater surface area? Please clarify.

L 72-75. The objective is vague and not clear. The authors should state the objective similar to what is indicated in the abstract section.

L 83Unless this reviewer missed something, the total number of animals in the tables and this section is 282, whereas the total number of animals described in L 104-107 is 216. Please clarify.

L 88: Delete “…at the age of about…”

L 91: Delete “…the time of..”

L 92: Delete “on” before a total

L 95: Delete “to each animal” and replace it with “24 h a day”

L 95-95. Please change to "cows were milked twice a day and delete the sentence that starts with “ad libitum. This has already been indicated in Line 95.

Could the authors explain why the dose of GnRH was so high (normal cows received 100 ug of GnRH) and the PGF dose was so low? It is not clear what kind of PGF was used. Was is it cloprostenol or was it dinoprost? Do authors know if this 0.5 mg PGF is effective in causing luteolysis?

Statistical Analysis

The experiment is designed unbalanced (216 cows in MOVS and only 68 cows in OVS). While this reviewer speculates why that is, It would be helpful for authors to explain the reason for this unbalanced design.

Please explain the final statistical model in your ANOCA analysis. First, as stated in L 137-138, it seems that the independent variable (the fix effects) in the model only included the season, protocol, and parity with no interactions. Second, based on the results in Tables 1 and 2, it appears the authors analyze the data using two separate models, a) examining the effect of season and protocol. Both tables shoed interaction effects, but this has not been mentioned in L 136-137. Why didn’t authors combine all the effects in one mode i.e. response variables = protocol, parity, parity X protocol, season, season x protocol, season X parity?

It appears that the authors have conducted some post-ANOVA mean separation procedures or mean contrasts. Please add this information to this section.

The reviewer questions the significant differences between the means in some of the variables. Provided the size of standard error in some variables means, including follicular diameter, interval to estrus, estrus duration, interval to estrus, are the authors certain that there was a significant effect of parity of protocols? As an example, the standard error for mean follicular diameters are between 1.6 to 2.8. across parities and protocols (Table 1). The raw mean for MOVS and OVS (regardless of parity) are 7.25 and 8.15 mm, respectively. Are they significantly different? If LS means for MOVS vs OVS or multiparous vs nulliparous are significantly different, due to the fact that the experiment was so unbalanced, then the authors MUST report these means in the body of the manuscript (please see the comment in the result section). Similarly, in Table 1 and Table 2, where there were main effects of season or protocol, without a significant interaction of the protocol by parity or protocol by season effects, the authors should report the main effect means within the body of the manuscript.  

Another example is the interval from 2nd GnRH to ovulation. The authors indicated that the was a P x PT interaction effect. There are no superscripts to show this significant interaction. With SE of 0.8 h, Is there a significant difference between 24.2 and 23.2 h?

The reviewer has similar concerns about the difference in some of the means and main effects significances in Table 2.

Please indicate what superscripts a and b mean in tables. The information has to be added in the tables legend

L 163: Change “higher” to “greater”

L 173: Please report the mean for summer. Longer interval in summer compared with what season? All other 3 seasons?? The mean interval to estrus in summer does not appear to that different from the mean interval in Autumn. be different.

In general, when there are significant interactions(which is the case in many measured variables in Tables 1 and 2), the authors need to be careful about how they interpret the data. For example, in L 176-178, the authors indicate that there is a protocol effect on estrus rate (i.e. estrus incidence), but no season effect. Thereafter, the authors continue saying that there was a season by protocol interaction effect on estrus rate. This means that the effect of protocol on estrus rate is not consistent and varies depending on the season. Overall, the 

the notion that protocol has a significant effect on several variables (follicular diameter, estrus response, estrus rate silent estrus, interval to ovulation, ovulation rate) has to be interpreted VERY carefully, given that authors detected significant interactions effects on the above-mentioned variables .

Discussion: (line numbering was missing in this section)

This reviewer disagrees with this interpretation of the data. MOVES effects on estrus response and ovulation rate were not significant (Table 1, P = 0.14 and P=0.10, respectively). Second, although the estrus response and ovulation rate were affected by the protocol in Table 2, the authors simply ignore the season-by-protocol interaction effect on these two variables (P= 0.004, and P=.007, respectively) One cannot, and should not, make such conclusions when the protocol effect across the season or parity is not consistent. Third, what do the authors mean MOVES is advantageous over OVS in improving fertility variable (6th line in the conclusion). Which fertility variable? The pregnancy rate was not different between the protocols.

1st Paragraph Discussion. How this data is useful to reduce the puberty age. What this experiment has to do with age to puberty. This a very broad and loose speculation. At best, the authors may conclude that the information may be useful in reducing the age at first calving!?

Authors claim that MOVS was a beneficial regimen to improves pregnancy during peak autumn ..... The authors' data does not support that. There was no effect of protocol on the pregnancy rate. This statement is inaccurate

Reviewer 2 Report

Dear Authors,

in my opinion the text is not easy to read, may be an editing of the English  language could be useful

But this is not the main problem that made me stop the reviewing process.
Actually I cannot understand the groups of animals involved in the experiment.

20          and also effects of season are also prominent

28          crossbred (swap × riverine)

88          were at age of about

89          with moderate body condition score

90-91    whereas, selected multiparous buffaloes had 2.5-3.5 BCS, 5-8 years old with an average BW of 588±89kg at the time of beginning of the experiment

92          The animals were fed on a total mixed ration (TMR) consisting of forage

93-94    (corn; 38%, soybean meal; 16%, linen; 6.0%, cottonseed cake; 6%,corn meal; 17.5%,vinasse; 10%,little su; 0.5% and premixed material; 6%).

95-97    Fresh and clean water was accessible 24h to each animal. Milking was practiced using a milking machine twice a day (6:00 and 18:00). Ad libitum fresh and clean water was provided routinely.

107        Buffaloes were synchronized either using MOVS

Reviewer 3 Report

General comments:

The manuscript “Effect of Modified and Conventional Ovsynch Regimens on Reproductive Variables of Crossbred (Swamp × Riverine) Nulliparous and Multiparous Buffaloes during Different Seasons” (Manuscript number; animals-1383684) was describes the effect of modified Ovsynch (MOVS) on fertility, estrus activity and ovarian dynamics.  

This manuscript contains some problems. First of all, why you treated “mifepristone” simultaneously  with 2nd GnRH treatment at Day 9? There is no explanation about the pharmacodynamic action and the role of the mifepristone in MOVS. Please explain in the introduction for easily understanding for readers. Second, why you treated hCG after artificial insemination simultaneously? In this experimental model, we could not clarify which treatment (mifepristone or hCG) was affected the pregnancy rate. Please describe the objectives of each treatment in “Introduction”, respectively Third, you described “In conclusion, incorporation of mifepristone and hCG in Ovsynch protocol enhances the fertility rates in heifers and multiparous buffaloes.” in the “5. Conclusions” section. However, I could not find the efficiency of MOVS on “Pregnancy rete” in Nulliparous buffaloes. In addition, from the analysis of logistic regression, there are no significant effect of “Protocol” and “Interaction of Parity and Protocol”. Therefore, you could not state the goodness of MOVS treatment. Also, what statistical test was used for comparing the Pregnancy rate of MOVS and OVS in Multiparous buffalos? Please describe it in the section of “Statistical analysis”.